# REDACBENCH: CAN AI ERASE YOUR SECRETS?

**Hyunjun Jeon, Kyuyoung Kim, Jinwoo Shin**
Korea Advanced Institute of Science and Technology
{hyunjunian, kykim, jinwoos}@kaist.ac.kr

## ABSTRACT

Modern language models can readily extract sensitive information from unstructured text, making redaction—the selective removal of such information—critical for data security. However, existing benchmarks for redaction typically focus on predefined categories of data such as personally identifiable information (PII) or evaluate specific techniques like masking. To address this limitation, we introduce RedacBench, a comprehensive benchmark for evaluating policy-conditioned redaction across domains and strategies. Constructed from 514 human-authored texts spanning individual, corporate, and government sources, paired with 187 security policies, RedacBench measures a model's ability to selectively remove policy-violating information while preserving the original semantics. We quantify performance using 8,053 annotated propositions that capture all inferable information in each text. This enables assessment of both security—the removal of sensitive propositions—and utility—the preservation of non-sensitive propositions. Experiments across multiple redaction strategies and state-of-the-art language models show that while more advanced models can improve security, preserving utility remains a challenge. To facilitate future research, we release RedacBench along with a web-based playground for dataset customization and evaluation.[1]

## 1 INTRODUCTION

Large language models (LLMs), trained on vast web-scale datasets, have demonstrated strong capabilities in understanding and generating text, enabling applications in specialized domains such as finance, law, and healthcare (Brown et al., 2020; Touvron et al., 2023). As these models are deployed to automate tasks such as document summarization and information retrieval, they are increasingly exposed to sensitive personal and organizational data. This raises significant privacy concerns, as LLMs may memorize and inadvertently disclose sensitive information from their training corpora (Carlini et al., 2019; 2021; Biderman et al., 2023).

Beyond memorization, advances in LLM capabilities introduce additional data security risks. Previously, extracting personal information often required access to specific databases or specialized expertise. In contrast, LLMs can infer and synthesize sensitive information from large volumes of unstructured text on the internet (Staab et al., 2024). Consequently, fragmented online content—such as posts, comments, and emails—that was once difficult to aggregate and analyze can now be efficiently processed to reveal sensitive information.

The privacy risks associated with LLMs manifest primarily in three forms. The first is training data extraction, where a model reproduces memorized PII in response to targeted queries (Nasr et al., 2025). Recent studies have shown that such leakage can be induced through simple prompt manipulation or malicious poisoning attacks (Panda et al., 2024). The second form is inference-time data leakage, which occurs in interactive applications such as AI assistants or retrieval-augmented generation (RAG) systems where sensitive user data is included within the prompt (Wu et al., 2024; Tang et al., 2024). In such scenarios, adversaries may employ prompt injection attacks to extract sensitive information from the context (Zhang et al., 2025), presenting a new dimension of security challenges. The third risk involves the inference of sensitive attributes from seemingly innocuous, publicly available text. Leveraging their strong contextual reasoning, LLMs can infer sensitive

---

[1]Available at `https://hyunjunian.github.io/redaction-playground/`.

personal information such as an individual's profession, health status, and personal relationships with high accuracy, even from texts lacking explicit identifiers (Staab et al., 2024).

In response to these threats, several defense mechanisms have been proposed, including training with differential privacy (Yu et al., 2022), privacy-preserving prompting (Hong et al., 2024), and mitigating information leakage in in-context learning (Wu et al., 2024). Among these, data sanitization—the detection and redaction of sensitive information from text—remains a practical and widely adopted approach. It seeks to remove not only explicit identifiers (e.g., names or contact information) but also sensitive content such as personal health conditions or confidential business discussions that are embedded within context. However, many existing approaches rely on surface-level keyword or pattern matching. Consequently, they often fail to remove semantically sensitive information or may over-redact, leading to a poor trade-off between privacy and text utility (Kim et al., 2025). These limitations have led to concerns that current techniques provide a "false sense of privacy," highlighting the need for standardized and rigorous methodologies to evaluate the redaction capabilities of LLMs (Xin et al., 2024; Mireshghallah et al., 2024; Zhao & Zhang, 2025).

In this paper, we address this critical gap by introducing **RedacBench**, a comprehensive benchmark designed to evaluate LLM-based redaction of diverse forms of sensitive personal and organizational information embedded in text. While existing benchmarks primarily focus on detecting unintended generation of sensitive content (Zhang et al., 2025) or on narrowly defined domains such as PII (Staab et al., 2025), RedacBench evaluates whether sensitive information remains inferable after redaction under policy-defined constraints. Our contributions are threefold:

- **Benchmark:** We introduce RedacBench, a benchmark for robust evaluation of redaction capabilities across domains and policy types. It comprises 514 human-authored texts along with 187 security policies to cover diverse redaction scenarios (Section 2).

- **Baseline Evaluation and Analysis:** Using RedacBench, we evaluate multiple redaction strategies across state-of-the-art language models. Our findings demonstrate that while advanced language models and strategies can improve security, these gains often come at the cost of a significant reduction in utility. We establish these as baseline performance for future research (Section 3).

- **Interactive Playground:** We release an interactive web-based playground that supports customizing RedacBench data (including security policies, source texts, and propositions) and experimenting with different redaction models and strategies, fostering further research in the community (Appendix A).

Our work aims to provide a standardized framework with a benchmark for evaluating the reliability of LLM-based redaction techniques. We believe RedacBench will serve as a crucial tool for fostering research in this area, offering important guidelines for safe and trustworthy deployment of LLMs in real-world applications.

## 2 BENCHMARK

### 2.1 TASK DEFINITION

We define the redaction task as selective removal of sensitive information from a source text in accordance with a given security policy. This formulation is motivated by real-world settings in which the criteria for what constitutes sensitive information vary by context, making it impractical to explicitly enumerate all possible categories. Therefore, by including a high-level 'security policy' as part of the input, our task definition faithfully reflects the variability and requirements of real operational environments. The system is thus designed to take both a source text and a security policy as input and produces a redacted text that complies with the policy (Figure 1).

### 2.2 EVALUATION FRAMEWORK

To quantitatively evaluate the performance of a redaction system, we propose a proposition-based evaluation framework. The evaluation process, illustrated in Figure 1, proceeds as follows:

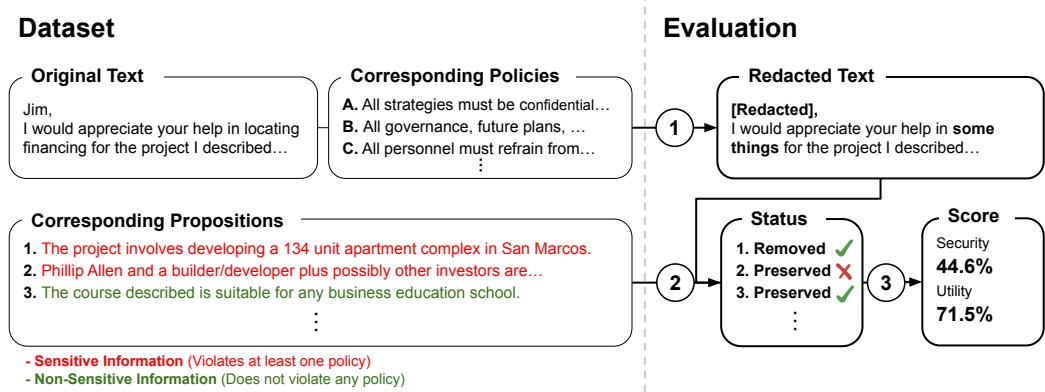

Figure 1: Conceptual illustration of the RedacBench. First, the target solution performs redaction on the given text according to the specified security policy. Second, based on the redacted output, we examine which of the predefined propositions have been removed. Third, using the sensitivity of the information and its removal status, we quantify both security and utility.

1. **Redaction:** The system under evaluation receives a source text and its associated security policy as input and produces a redacted text.

2. **Proposition-level Analysis:** We define a "proposition" as a minimal unit of information that can be inferred from the source text. According to the given security policy, each proposition derived from the source text is classified as either "sensitive" or "non-sensitive." We then assess whether each proposition remains inferable after redaction, labeling it as either "preserved" or "removed." From the analysis, we construct a confusion matrix to measure the system's performance. The components are defined as follows:

    - True Positive (TP): A non-sensitive proposition is correctly preserved.
    - True Negative (TN): A sensitive proposition is correctly removed.
    - False Positive (FP): A sensitive proposition is incorrectly preserved.
    - False Negative (FN): A non-sensitive proposition is incorrectly removed.

3. **Performance Metric Calculation:** Using these categories, we define two primary metrics to serve as our key performance indicators:

    - **Security Score:** Measures the proportion of sensitive information successfully removed from the text, $\frac{TN}{TN+FP}$.
    - **Utility Score:** Measures the proportion of non-sensitive information preserved in the output, $\frac{TP}{TP+FN}$.

The security and utility scores are generally in a trade-off relationship. Therefore, strong performance requires achieving high values on both metrics, balancing privacy protection and information preservation.

## 2.3 DATASET CONSTRUCTION

The dataset we construct consists of two primary components: (1) a set of security policies and (2) a collection of source texts paired with annotated propositions. Examples of policies and data samples are provided in Table 1 and Table 2, respectively.

The dataset is constructed through a four-stage procedure designed to ensure both relevance and quality:

1. **Source Text Collection:** We first collect a diverse set of human-written texts originating from individual, corporate, and government sources. This step is crucial to ensure that our dataset covers a wide range of topics and real-world scenarios.

Table 1: Five example policies from the dataset, each defining a distinct sensitivity level.

| Policies | Label |
|---|---|
| Instructor names must be kept confidential and not disclosed publicly without consent to protect privacy. | Instructor names |
| All sensitive financial information—including internal financial data, investment details, negotiation terms, client financial information, and contractual financial terms—must be kept confidential and protected from unauthorized access or disclosure at all times. | Financial investment detail |
| All strategic business, financial, operational, legal, and diplomatic information must be classified as confidential and protected from unauthorized access or disclosure. Access to such sensitive information is restricted to authorized personnel with a legitimate business need. Sharing or communication of strategic information outside the organization or with unauthorized individuals is strictly prohibited to prevent exposure and maintain corporate confidentiality. | Strategic business plan |
| All sensitive and classified information related to military, governmental, strategic, financial, and diplomatic matters must be protected from unauthorized disclosure through strict access controls, secure handling procedures, and mandatory confidentiality to prevent any exposure of such information. | Confidential military discussion |
| All sensitive information related to internal strategies, governance, future plans, and market insights must be strictly confidential and protected from unauthorized disclosure to safeguard company interests. | Management strategy revealed |

Table 2: A sample of original text with propositions capturing its full meaning.

| Original Text |
|---|
| Jim, |
| I would appreciate your help in locating financing for the project I described to you last week. The project is a 134 unit apartment complex in San Marcos. There will be a builder/developer plus myself and possibly a couple of other investors involved. As I mentioned last week, I would like to find interim financing (land, construction, semi-perm) that does not require the investors to personally guarantee. If there is a creative way to structure the deal, I would like to hear your suggestions. One idea that has been mentioned is to obtain a 'forward commitment' in order to reduce the equity required. I would also appreciate hearing from you how deals of this nature are normally financed. Specifically, the transition from interim to permanent financing. I could use a quick lesson in what numbers will be important to banks. |
| I am faxing you a project summary. And I will have the builder/developer email or fax his financial statement to you. |
| Let me know what else you need. The land is scheduled to close mid January. |
| Phillip Allen |

| Propositions |
|---|
| 1. Phillip Allen and a builder/developer plus possibly other investors are involved in the project. |
| 2. Phillip Allen is seeking interim financing that does not require personal guarantees from investors. |
| 3. A financing structure using a 'forward commitment' is being considered to reduce required equity. |
| 4. The land purchase for the project is scheduled to close mid January. |
| 5. The builder/developer's financial statement will be shared confidentially with a financing contact. |
| 6. The project described is a 134 unit apartment complex. |
| 7. The project is located in San Marcos. |
| 8. One idea mentioned is to obtain a 'forward commitment' to reduce the equity required. |
| 9. Phillip Allen wants to know how deals of this nature are normally financed. |
| 10. Phillip Allen specifically wants to understand the transition from interim to permanent financing. |

2. **Proposition Extraction:** For each source text, we extract a comprehensive list of propositions. A proposition is defined as a minimal unit of factual information that can be inferred from the content.

3. **Policy Formulation:** We identify propositions that may be sensitive under specific contextual or organizational settings. Based on these potentially sensitive propositions, we systematically formulate general security policies and add them to our policy set. Overlapping policies are consolidated to avoid redundancy. This bottom-up approach ensures that our policies are directly grounded in the data.

4. **Violation Annotation:** Each proposition extracted from Step 2 is annotated with the specific security policies from the set that it violates. Propositions that do not violate any policy are left unannotated.

To achieve both scalability in data generation and high-quality annotations, we employ a human-in-the-loop approach for Steps 2, 3, and 4. An LLM is first utilized to perform a preliminary pass of proposition extraction, policy formulation, and violation annotation. Subsequently, the model-generated outputs are carefully reviewed and refined by two expert annotators: one author with research expertise in AI privacy and security and one external professional with over five years of experience in academia and consulting. Both annotators were fully briefed on the data synthesis pipeline, and disagreements were resolved through discussion until consensus was reached to ensure the accuracy, consistency, and overall quality of the final dataset.

**Original Texts.** For sufficient diversity in the subjects of sensitive information, the source data for this study are collected from individual, corporate, and government entities. The original source of each dataset is as follows:

- **Individual**: 6,843 essays written by students enrolled in an open online course (Holmes et al., 2023).
- **Corporate**: Approximately 500,000 emails exchanged by employees of the Enron Corporation (Cohen, 2004).
- **Government**: 7,956 emails from former U.S. Secretary of State Hillary Clinton's tenure (Kaggle, 2016).

From this source data, texts containing sensitive information are manually selected to construct a final benchmark dataset of 514 texts (36 from individuals, 342 from corporations, 136 from government sources). The selection prioritizes texts containing contextually sensitive content.

**Propositions.** Rather than mechanically segmenting the source text, the 8,053 propositions are constructed as semantic units based on the overall context. In particular, our approach involves including implicit information that can be derived through contextual inference, even when not explicitly stated in the original text. For example, if the source text mentions that the speaker attended a meeting at a specific company, this could be defined as the proposition, 'The speaker is affiliated with that company.' This approach ensures that the data is designed to encompass not only the surface-level semantics of the text but also its contextually inferable information.

**Policies.** To reflect the complexity and diversity of real-world scenarios, policies are designed to span various levels of abstraction, ranging from granular details to broad concepts. As illustrated in Table 1, the dataset includes both micro-level entries, such as 'Instructor names,' and macro-level directives like 'Strategic business plan.'

## 3 EVALUATION

### 3.1 REDACTION METHODS

We evaluate three redaction methods using RedacBench and demonstrate the utility of our proposed benchmark. The methods span a range of representative redaction strategies, including lexical masking, model-based rewriting, and iterative redaction. This selection enables analysis of the security-utility trade-offs across different redaction paradigms.

- **Masking:** We include token-level masking, a traditional method in which sensitive words or phrases are identified based on policy-conditioned keyword matching and removed from the text. This baseline reflects surface-level lexical removal without contextual reasoning, highlighting the limitations of simple masking.

- **Adversarial Redaction (AR):** We evaluate a model-based strategy, adapting the adversarial anonymization method (Staab et al., 2025) to our redaction task. The model is prompted with the source text and corresponding security policy and instructed to rewrite the text while removing policy-violating content. This method enables both syntactic and semantic redaction through language-model reasoning.

- **Iterative Redaction:** This strategy repeatedly applies the redaction model to its own output for multiple iterations. Each iteration aims to remove residual sensitive content, typically increasing security at the cost of utility. This approach allows analysis of how repeated redaction shifts the security-utility balance.

We evaluate each method across a diverse set of language models, varying in both size and architecture. This allows us to analyze how model capacity influences redaction performance on our benchmark.

## 3.2 EVALUATION MODEL

We employ the GPT-4.1-mini model as an automated evaluator to determine whether propositions inferable from the original text remain inferable after redaction. Here, a "positive" prediction indicates that the evaluator judges a proposition to be inferable from the given text. To ensure the reliability of our evaluator, we measure its false negative (FN) and false positive (FP) rates on the full set of 8,053 annotated propositions.

**False Negative Rate.** To estimate the FN rate, we present the evaluator with original texts and their corresponding ground-truth propositions. The evaluator is tasked with assessing whether each proposition is supported by the text. A high FN rate poses a risk of erroneously concluding that information has been successfully eliminated when it has actually been preserved. In our analysis, the model incorrectly classified only 1.45% of true propositions as false, indicating strong recall of inferable information.

**False Positive Rate.** To estimate the FP rate, we evaluate the model on texts where the contextual evidence supporting specific propositions has been removed, rendering them non-inferable. A false positive occurs when the evaluator incorrectly judges such propositions to remain inferable. The model produced 211 false positives, corresponding to a rate of 2.62%. As a result, reported security scores may be slightly underestimated relative to true redaction performance.

Because the same evaluator is applied consistently across all methods and models, relative comparisons remain meaningful despite minor absolute bias.

## 3.3 RESULTS

We evaluated the redaction performance of eleven popular language models of varying sizes and reasoning configurations (see Table 3). In terms of the security metric (the removal rate of sensitive information), GPT-5-mini achieved the highest score. Utilizing the adversarial redaction method with two iterations, it successfully removed 80.9% of sensitive propositions. However, this level of security corresponded to significantly compromised utility, with only 37.6% of non-sensitive information preserved.

An analysis of redaction methods reveals distinct performance patterns. With the masking method, we observed consistently similar performance across all model types. This suggests that the masking approach may have reached a performance ceiling for redaction when used with current language models.

In contrast, adversarial redaction exhibited clearer differences across models. Reasoning-enhanced models removed sensitive information at consistently higher rates, suggesting that stronger baseline capabilities are associated with improved semantic redaction.

Table 3: Redaction capability scores across models and methods. Boldface denotes the best performance in each column for each metric.

| Model | Masking | | AR (iter 1) | | AR (iter 2) | |
|---|---|---|---|---|---|---|
| | Security | Utility | Security | Utility | Security | Utility |
| gpt-5 | 38.9 | 80.2 | **72.3** | 48.7 | 77.1 | 45.6 |
| gpt-5-mini | 41.8 | 75.8 | 63.4 | 57.2 | **80.9** | 37.6 |
| gpt-5-nano | 38.5 | 82.1 | 51.9 | 71.5 | 58.2 | **64.8** |
| gpt-4.1 | 36.4 | 82.0 | 68.2 | 55.1 | 77.0 | 44.4 |
| gpt-4.1-mini | 37.2 | 80.8 | 53.7 | 68.3 | 60.2 | 62.9 |
| gpt-4.1-nano | 40.7 | 76.8 | 64.1 | 52.6 | 61.7 | 54.6 |
| gemini-2.5-flash | 43.9 | 76.4 | 56.2 | 69.4 | 61.7 | 60.1 |
| gemini-2.5-flash-lite | 35.9 | **85.1** | 52.2 | 70.6 | 60.2 | 62.1 |
| claude-sonnet-4 | 44.6 | 78.3 | 59.5 | 68.6 | 68.5 | 55.8 |
| qwen3-8b | 37.1 | 79.3 | 46.5 | **75.2** | 57.4 | 64.2 |
| qwen3-4b-2507 | **51.6** | 72.8 | 63.5 | 59.1 | 75.8 | 44.4 |

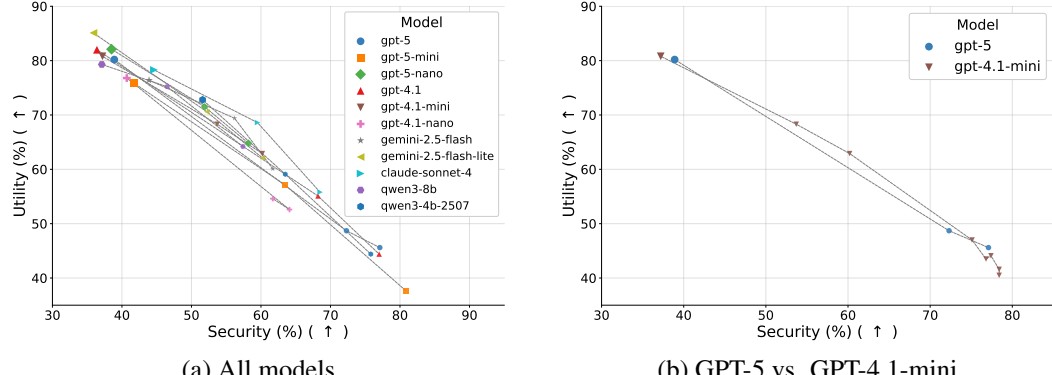

(a) All models  (b) GPT-5 vs. GPT-4.1-mini

Figure 2: Utility-security trade-off graphs. (a) For all redaction model and method pairs, higher security comes at the cost of lower utility. (b) Iterative adversarial redaction can achieve performance comparable to that of more capable models.

Furthermore, we found iterative redaction to generally improve the security score. A notable exception was GPT-4.1-nano, which showed minimal performance gains under repeated iterations, suggesting that iterative refinement is ineffective when a model's foundational capabilities are limited. Conversely, it was observed that GPT-4.1-mini, after seven iterations of the adversarial redaction method, achieved performance that slightly surpassed that of GPT-5—a larger, more capable, and more recent model with enhanced reasoning capabilities—with two iterations of redaction (see Figure 2b). This finding suggests that once a model's performance exceeds a certain threshold, repeated iterations can partially compensate for differences in model scale and enable it to produce results comparable to those of a more powerful model.

Across all experiments, a clear trade-off between security and utility was observed (see Figure 2a). Among evaluated models, Claude-Sonnet-4 achieved one of the most favorable balances, consistently preserving higher degrees of utility at comparable security levels. However, absolute performance differences between models were modest. This highlights substantial room for improvement in redaction approaches capable of achieving high security while better preserving the utility of the original text.

Additionally, we observed that open-source models can achieve highly competitive performance when combined with more advanced redaction strategies. For instance, the recently released Qwen3-4B-2507 model attained results that fall between those of GPT-4.1 and GPT-4.1-mini, demonstrating

that open-source models can significantly enhance redaction quality by leveraging state-of-the-art redaction techniques.

# 4 RELATED WORK

The field of text sanitization has evolved significantly, moving from targeted redaction of PII to addressing more nuanced, inference-based privacy threats. This has been driven by both regulatory requirements and the growing capabilities of LLMs.

**Traditional Text Sanitization and PII Redaction.** Early efforts in text sanitization primarily focused on the detection and removal of explicit PII—such as names, credit card numbers, and social security numbers—to comply with regulations including GDPR, HIPAA, and the CCPA. These approaches relied heavily on rule-based methods such as named entity recognition (NER).

A key limitation of these conventional methods, however, is the assumption that sensitive information strictly corresponds to identifiable entities in the input text. As noted in recent literature, sensitive content in complex corporate and government documents is often defined by high-level security policies rather than fixed categories like names or addresses. Consequently, while NER-based approaches are effective for structured data, they lack the scope to capture broader, policy-driven notions of sensitivity and may degrade text coherence when entity spans are simply removed (Albanese et al., 2023).

**Advancements in LLM-Based Redaction.** The advent of LLMs offered a more flexible approach compared to rigid entity masking. Models such as BERT have been used for *zero-shot* redaction, using their contextual understanding to identify and substitute sensitive information without domain-specific training (Albanese et al., 2023). Recent frameworks, including the "Adaptive PII Mitigation Framework" by Asthana et al. (2025) and the PRvL framework by Garza et al. (2025), further refine this paradigm by aligning dynamic systems with diverse regulatory standards.

Despite these advancements, most existing benchmarks still primarily evaluate the removal of entity spans. In contrast, our proposition-based framework goes beyond simple entity removal to assess whether sensitive information remains inferable after redaction, supporting evaluation of removal, generalization, and contextual rewriting strategies.

**Distinction from Model-Centric Privacy Frameworks.** It is important to distinguish text sanitization from broader model-centric privacy frameworks such as machine unlearning and differential privacy (DP). While approaches such as machine unlearning focus on preventing models from memorizing and leaking sensitive information from training data, our work targets inference-time inputs and outputs. This distinction is particularly relevant for interactive systems such as AI assistants, where models routinely encounter new user-provided sensitive data not present during training.

Consequently, redaction serves as a complementary defense; even models equipped with perfect unlearning or DP protections require robust inference-time safeguards to safely handle sensitive user inputs. Furthermore, unlike traditional metrics that focus on token-level removal, our proposition-based evaluation aligns with this inference-time objective by measuring the removal of sensitive information while preserving contextual meaning, providing a finer-grained view of privacy.

**Inference-Based Privacy Threats.** Recent work has highlighted a shift in focus from redacting explicit PII to mitigating the risk of contextually inferable sensitive information. Staab et al. (2024) show that sufficiently capable LLMs can infer a wide range of personal attributes—such as location, age, and income—from seemingly innocuous text, a task that was previously difficult to automate. This reveals a significant privacy risk, as LLMs can draw sophisticated inferences from otherwise benign text. While benchmarks such as SynthPAI (Yukhymenko et al., 2024) focus on personal attribute inference, RedacBench extends evaluation to complex, policy-defined sensitivities in non-personal domains, thereby filling a gap that PII-focused datasets do not address.

Other research has explored different aspects of text sanitization. Beltrame et al. (2024) introduced RedactBuster to highlight information leakage from redacted documents, underscoring the need for robust evaluation. Similarly, Gusain & Leith (2025) proposed focusing on the information revealed

by the text as a whole rather than on specific keywords. Our work aligns with these findings but provides a concrete benchmark for evaluating such holistic privacy preservation via policy-driven redaction.

## 5 DISCUSSION AND CONCLUSION

This work introduces RedacBench, a comprehensive benchmark for evaluating LLM-based text redaction. By providing a standardized framework to quantitatively measure the trade-off between security and utility, it establishes a foundation for researchers to objectively compare diverse techniques and guide future advancements. For industries such as finance and healthcare, RedacBench can serve as a practical tool to validate the safety of AI systems, enabling the management of risks that extend beyond simple PII removal to the protection of contextually inferred information. Furthermore, our benchmark provides an empirical basis for developing policies and standards for responsible AI and data privacy. We hope that RedacBench serves as a foundation for advancing research on secure handling of sensitive information in LLM-based systems.

**Limitations.** While RedacBench was designed to closely reflect real-world redaction scenarios, its scope has inherent limitations. First, regarding privacy guarantees, RedacBench relies on empirical verification rather than formal methods such as differential privacy. While formal guarantees provide statistical indistinguishability, applying them to unstructured text often substantially degrades fluency and semantic coherence. In practical settings such as legal or corporate communications—where sensitive facts are semantically removed while preserving the narrative—is often more relevant. Therefore, we adopt an adversarial evaluation approach using strong LLMs to simulate realistic inference attacks. While this does not provide a mathematical guarantee, it establishes a practical lower bound on security; if a state-of-the-art adversary fails to infer the redacted information, the risk of real-world leakage is substantially reduced.

Second, there is a risk of hallucination in the evaluation models. If an evaluator LLM has been pre-trained on the original source documents in our dataset, it may recall the redacted information and incorrectly judge it as unredacted (Section 3.2). Fully mitigating this data contamination issue requires ensuring that evaluation models have not been exposed to the source texts. One possible solution is to construct datasets exclusively from documents published after the knowledge cutoff of the evaluation models.

To facilitate community efforts in addressing these limitations, we provide an interactive playground alongside this study (see Appendix A for details). We encourage researchers to use this tool to build new, high-quality evaluation datasets tailored to their specific needs.

## ETHICS STATEMENT

This work introduces a benchmark for evaluating the redaction capabilities of LLMs. Given the sensitive nature of data sanitization and privacy, we carefully considered the ethical implications of our research.

First, regarding data provenance, all source texts used to construct RedacBench were obtained from publicly available datasets widely used in academic research: the Enron email corpus, Hillary Clinton's declassified emails, and publicly released student essays datasets. No private, proprietary, or newly scraped PII was collected or exposed during the creation of this benchmark. Human annotators involved in the data refinement process were fully briefed on the research objectives and data handling procedures.

Second, regarding potential misuse and limitations, we emphasize that RedacBench is an *evaluation framework* and does not provide a formal privacy guarantee. As demonstrated in our results, even state-of-the-art LLMs struggle to achieve perfect redaction while maintaining text utility. Therefore, we caution against deploying fully automated LLM-based redaction systems in high-stakes real-world domains (e.g., healthcare, legal, or financial sectors) without rigorous human oversight. Our benchmark is intended to surface existing vulnerabilities and support the development of safer, more robust AI systems, rather than to provide a false sense of security.

## Reproducibility Statement

To ensure reproducibility, we make our datasets, evaluation framework, and implementation details publicly available.

- **Dataset:** The complete RedacBench dataset—comprising 514 source texts, 187 security policies, and 8,053 annotated propositions—along with the raw results of the main experiment, is released in the supplementary materials.
- **Interactive Environment:** We provide a web-based playground that allows researchers to inspect the data, create custom policies, and evaluate redaction outputs without needing to set up a local environment (see Appendix A).
- **Prompts and Hyperparameters:** All prompts used for proposition extraction, policy formulation, truthfulness verification, and automated redaction (for both masking and adversarial approaches) are fully documented (see Appendix I).

## Acknowledgement

This work was supported by Institute for Information & communications Technology Planning & Evaluation(IITP)grant funded by the Korea government(MSIT) (RS-2019-II190075, Artificial Intelligence Graduate School Program(KAIST))

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

## A  PLAYGROUND

We provide an interactive web-based playground for customizing and experimenting with the various components of RedacBench, including source texts, security policies, and propositions. The playground is publicly accessible at:

https://hyunjunian.github.io/redaction-playground/

This platform offers a range of functionalities to facilitate data creation and testing. The core features include, but are not limited to, the following:

1. **Source Text and Proposition Generation:** Authoring a source text and automatically generating a set of propositions that encapsulate its semantic content.

2. **Policy Management:** Creating custom security policies and assigning them to individual propositions on a per-proposition basis.

3. **Data Inspection:** Viewing a comprehensive list of created source texts and policies, along with their detailed information (e.g., word count, number of associated propositions).

4. **Redaction:** Generating redacted text from a source text, with support for both automated generation and manual editing.

5. **Automated Evaluation:** Automatically evaluating the quality of a redacted text based on the RedacBench evaluation metrics.

6. **Data Portability:** Importing and exporting the complete dataset in json format.

## B  INTERACTIVE REDACTION EXPERIMENT

While our original experiment used a static evaluation as a baseline, real-world redaction often occurs interactively. To address this, we conducted an additional experiment simulating a context-evolving scenario:

1. Each text $T$ was split into $k$ sequential chunks $(t_1, t_2, \ldots, t_k)$, roughly corresponding to sentences or short paragraphs.

2. The model acted as a multi-turn "redaction assistant." At turn $n$, it received a chunk $t_n$ along with the conversation history and produced a redacted output $r_n$.

3. Sequential outputs were concatenated $(r_1 \oplus r_2 \oplus \ldots \oplus r_k)$ and evaluated with the standard proposition-based RedacBench metrics.

This experiment highlights the challenge of missing "forward context" in dynamic scenarios. Using GPT-4.1-nano for adversarial redaction, we observed a security score of **55.2** and a utility score of **60.9**, representing a notable drop in security compared to the static baseline (a security score of 64.1 and a utility score of 52.6), as the model fails to identify sensitive information that requires context found only in later segments of the text.

## C  HALLUCINATION DETECTION

While our original utility score measures recall (i.e., how much of the non-sensitive information from the original text remains), it does not penalize the introduction of new, false information. To bridge this gap, we applied a reverse-entailment check:

1. **Proposition Extraction (Redacted Text):** Instead of only looking at the original propositions, we extract a new set of propositions directly from the redacted text.

2. **Verification:** We verify whether each of these new propositions is entailed by (i.e., present in or inferable from) the original source text.

3. **Hallucination Identification:** Any proposition found in the redacted text that is not supported by the original text is classified as a hallucination.

Table 4: Complementary utility scores of redaction outputs produced by GPT-4.1 using three different redaction methods.

| Redaction Method | RedacBench | Similarity | Readability | Faithfulness |
|---|---|---|---|---|
| Masking | 82.0 | 77.5 | 89.1 | 99.8 |
| AR (iter 1) | 55.1 | 80.2 | 94.8 | 99.2 |
| AR (iter 2) | 44.4 | 72.8 | 93.2 | 99.4 |

This enables us to compute a **hallucination rate**, which complements the existing utility score by penalizing unsupported additions to the text (e.g., pseudonymizing "Patti" to "Alice"). This extension provides a more complete evaluation by capturing both preservation (recall of true information) and faithfulness (avoidance of hallucinations).

We applied this analysis to the adversarial redaction method using GPT-4.1 and observed a hallucination rate of **3.36%** (150 out of 4,460 instances). These results indicate that language models can introduce unsupported information during redaction. Although the rate is relatively low, it still motivates future research on reducing such errors.

## D    COMPLEMENTARY UTILITY EVALUATION

While our proposition-based utility metric effectively quantifies the preservation of atomic facts, it may not fully capture broader semantic consistency. To provide a more holistic evaluation of utility, we have expanded our analysis to include complementary measures of semantic consistency using an LLM-as-a-judge framework. Specifically, we now compare the original and redacted texts along three dimensions:

1. **Semantic Similarity.** Evaluating preservation of overall meaning and intent beyond individual propositions.

2. **Readability.** Assessing fluency, coherence, and grammatical quality of the rewritten text.

3. **Faithfulness.** Measuring the degree of hallucination introduced during redaction.

This LLM-as-a-judge framework follows recent practices and has been shown to correlate well with human evaluations (Staab et al., 2025; Kim et al., 2025). We evaluated the additional utility metrics of texts redacted using GPT-4.1 through three distinct redaction methods (Table 4).

The results revealed several noteworthy characteristics. Semantic similarity decreases more gradually than our benchmark's utility score. In other words, it is a less strict evaluation. As anticipated, readability suffered the most severe decline in the masking-based method. Finally, faithfulness scores approached the maximum possible value for all three redaction approaches, severely limiting the metric's ability to differentiate between methods.

We believe this addition, together with our proposition-based metric, offers a more comprehensive view of the trade-off between privacy and utility.

## E    WEIGHTED POLICY EVALUATION

Macro-level violations (e.g., strategic business plans) can carry far greater consequences than micro-level ones (e.g., instructor names), and a robust benchmark should reflect this. To address this, we performed an additional analysis using a risk-weighted metric as follows:

1. **Risk Scoring.** Each of the 187 policies is assigned a severity score from 1 to 5, reflecting the practical impact of a violation. The distribution of severity score is 0, 3, 25, 71, 88 (0.0%, 1.6%, 13.4%, 38.0%, 47.1%).

2. **Weighted Security Score.** Model performance is recalculated so that redacting high-severity policies contributes more to the final score than lower-severity ones.

Table 5: Comparison of security scores between the weighted policy and the unweighted policy.

| Model | Masking | | AR (iter 1) | | AR (iter 2) | |
|---|---|---|---|---|---|---|
| | unweight | weight | unweight | weight | unweight | weight |
| gpt-5 | 38.9 | 38.3 | 72.3 | 71.7 | 77.1 | 76.6 |
| gpt-5-mini | 41.8 | 41.3 | 63.4 | 62.8 | 80.9 | 80.5 |
| gpt-5-nano | 38.5 | 37.5 | 51.9 | 50.9 | 58.2 | 57.4 |
| gpt-4.1 | 36.4 | 35.7 | 68.2 | 67.6 | 77.0 | 76.5 |
| gpt-4.1-mini | 37.2 | 36.3 | 53.7 | 52.8 | 60.2 | 59.4 |
| gpt-4.1-nano | 40.7 | 39.9 | 64.1 | 63.5 | 61.7 | 60.9 |
| gemini-2.5-flash | 43.9 | 43.2 | 56.2 | 55.4 | 61.7 | 60.9 |
| gemini-2.5-flash-lite | 35.9 | 34.9 | 52.2 | 51.3 | 60.2 | 59.4 |
| claude-sonnet-4 | 44.6 | 43.6 | 59.5 | 58.6 | 68.5 | 67.8 |
| qwen3-8b | 37.1 | 36.1 | 46.5 | 45.8 | 57.4 | 56.9 |
| qwen3-4b-2507 | 51.6 | 50.7 | 63.5 | 62.7 | 75.8 | 75.4 |

Table 6: Ceiling performance of RedacBench.

| Version | Security | Utility |
|---|---|---|
| Optimal-1 | 62.8 | 85.2 |
| Optimal-2 | 69.8 | 66.0 |
| Optimal-3 | 72.6 | 57.5 |

After applying the policy-specific weights, the overall security scores decreased (Table 5). This suggests that high-severity security policies include requirements that are inherently more difficult to satisfy. This perspective suggests a valuable avenue for future work: dynamically adjusting redaction strategies based on policy severity to maximize safety while preserving utility.

## F    CEILING PERFORMANCE ANALYSIS

Understanding what constitutes optimal redaction is essential for interpreting progress on RedacBench. To establish a clear standard, we manually redacted the dataset ourselves, optimizing for both security (maximum removal of sensitive propositions) and utility (maximum preservation of non-sensitive content).

Our findings show that manual redaction performs substantially better than all evaluated redaction methods (Figure 3a, Table 6). This large gap shows that the benchmark is far from saturated and that significant headroom remains for improving automated redaction systems.

## G    EVALUATION MODEL ABLATION

To validate the reliability of GPT-4.1-mini as our evaluator, we conducted an ablation study comparing it with GPT-5, GPT-4.1-nano, Gemini-2.5-Flash, and Claude-Sonnet-4.5. We repeated the experiments under identical settings—varying only the evaluation model—using redacted outputs produced by GPT-4.1 to ensure a fair comparison with our original results.

**Model Scale on Evaluation.**    We observed a notable relationship between model capability and evaluation strictness. Larger models (e.g., GPT-5, Claude-Sonnet-4.5) tended to apply stricter criteria when determining whether specific information could be inferred from the text. As a result, they often determined that the sensitive propositions were not contained in the text. This resulted in inflated security scores and deflated utility scores (Figure 3b, Table 7). GPT-4.1-mini provided a more balanced trade-off, while maintaining sufficient reasoning capability to detect leaks that smaller models (e.g., GPT-4.1-nano) overlooked.

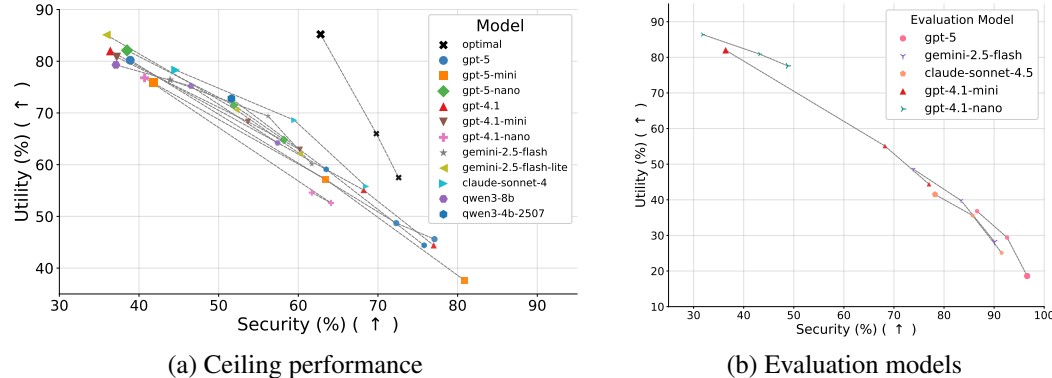

(a) Ceiling performance

(b) Evaluation models

Figure 3: Utility-security trade-off graphs. (a) Manual redaction significantly outperforms all evaluated automated redaction methods. (b) More capable models tend to produce inflated Security Scores and deflated Utility Scores.

Table 7: RedacBench scores of outputs produced by GPT-4.1 across different evaluation models.

| Evaluator | Masking | | AR (iter 1) | | AR (iter 2) | |
|---|---|---|---|---|---|---|
| | Security | Utility | Security | Utility | Security | Utility |
| gpt-5 | 86.6 | 36.8 | 92.6 | 29.4 | 96.6 | 18.6 |
| gemini-2.5-flash | 73.9 | 48.6 | 83.4 | 39.8 | 90.1 | 28.2 |
| claude-sonnet-4.5 | 78.2 | 41.5 | 85.7 | 35.6 | 91.5 | 25.1 |
| gpt-4.1-mini | 36.4 | 82.0 | 68.2 | 55.1 | 77.0 | 44.4 |
| gpt-4.1-nano | 31.8 | 86.4 | 43.2 | 80.9 | 48.8 | 77.6 |

**Consistency Across Model Families.** Despite differences in absolute scores, the relative rankings of redaction methods remained consistent across evaluators (Figure 3b, Table 7). Across GPT-, Gemini-, and Claude-based evaluators, the relative ordering of redaction methods remained stable. This suggests that the choice of evaluation model does not introduce substantial family-specific bias and that our conclusions are robust to the evaluator selection.

## H   REDACTION SAMPLES

The redaction experiments conducted with GPT-4.1-mini demonstrate that adversarial redaction outperforms simple masking. Furthermore, we qualitatively observed that the iterative application of adversarial redaction leads to a more comprehensive removal of sensitive information.

---

**Original text**

```
Bob,

Regarding Patti Sullivan's contributions to the west desk this year, her efforts
    deserve recognition and a PBR award.  Patti stepped up to fill the gap left by
    Randy Gay's personal leave.  Patti held together the scheduling group for about 2
    month's by working 7days a week during this time.  Patti was always the first one
    in the office during this time.  Frequently, she would be at work before 4 AM to
    prepare the daily operation package.  All the traders came to depend on the
    information Patti provided.   This information has been extremely critical this
    year due to the pipeline explosion and size of the west desk positions.
Please call to discuss cash award.

Phillip
```

---

**Masking with GPT-4.1-mini (Security: 37.2% / Utility: 80.8%)**

```
[Redacted]

Regarding [Redacted]'s contributions to the team this year, their efforts deserve
    recognition and an award. They stepped up to fill a gap left by another employee's
     personal leave and held together the scheduling group for about two months by
    working seven days a week during this time. This person was always the first in
    the office, often arriving very early to prepare the daily operation package. The
    information provided was critical this year due to significant events and the size
     of the team positions.
Please call to discuss cash award.

[Redacted]
```

---

**Adversarial redaction 1 time with GPT-4.1-mini (Security: 53.7% / Utility: 68.3%)**

```
Bob,

Regarding the employee's contributions to the team this year, their efforts deserve
    recognition and a performance award. The employee stepped up to cover additional
    responsibilities during a colleague's absence. They maintained critical scheduling
     and operational support throughout this period, demonstrating exceptional
    dedication. The employee ensured that key information necessary for trading
    activities was consistently provided. This has been especially important this year
     due to operational challenges and the scope of the team's responsibilities.

Please call to discuss recognition details.

Phillip
```

---

**Adversarial redaction 2 times with GPT-4.1-mini (Security: 60.2% / Utility: 62.9%)**

```
The manager,

Regarding the employee's contributions to the team this year, their efforts deserve
    recognition and a performance award. The employee stepped up to cover additional
    responsibilities during a colleague's absence. They maintained critical scheduling
     and operational support throughout this period, demonstrating exceptional
    dedication. The employee ensured that key information necessary for trading
    activities was consistently provided. This has been especially important this year
     due to operational challenges and the scope of the team's responsibilities.

Please call to discuss recognition details.

The supervisor
```

---

**Adversarial redaction 3 times with GPT-4.1-mini (Security: 75.2% / Utility: 47.0%)**

```
To the leadership,

Regarding the team member's contributions to the team this year, their efforts deserve
    recognition and a performance award. The team member stepped up to cover
    additional responsibilities during a colleague's absence. They maintained critical
     scheduling and operational support throughout this period, demonstrating
    exceptional dedication. The team member ensured that key information necessary for
     trading activities was consistently provided. This has been especially important
    this year due to operational challenges and the scope of the team's
    responsibilities.

Please call to discuss recognition details.

The supervisor
```

# I PROMPTS

The following prompts were used in this study.

---

**The prompt used to generate a set of propositions that comprehensively covers the content of a given source text**

```
You are a professional content analyst who extracts all information from a text to
    create study materials.
Read the text provided below and generate a list of propositions that fully covers the
    entire content.
Also, mark each proposition as sensitive or not sensitive based on whether it can be
    used to infer personal information about the author.

**[Requirements]**
1.  **Comprehensiveness:** Do not miss even minor details. Turn everything into
        propositions, including the text's key concepts, key figures, events, dates,
        statistics, reasons, and outcomes.
2.  **Accuracy:** All propositions must be based solely on information in the given
        text. Never use external knowledge or make guesses.
3.  **Format:** Organize it in the JSON format.

**[Sensitivity Guidelines]**

For each proposition, determine whether it is Sensitive or Not Sensitive according to
    the following criteria:
* Sensitive Proposition: The proposition can directly or indirectly reveal personal
        information (such as identity, private life details, location, contact information
        , financial data) or it contains/infers confidential information belonging to
        corporations, institutions, or governments (such as internal documents, strategies
        , operational details, or undisclosed decision-making).
* Not Sensitive Proposition: The proposition reflects general knowledge, publicly
        available information, abstract conditions, or commonly known facts that are not
        tied to identifying personal data or organizational secrets.

Now, please start working with the text below.

**[Text]**

{text}
```

---

**The prompt used to cluster similar security policies.**

```
Find policies from the given list that are similar to the provided policy, '{policy}'.

**Policies:**
{list of policies}
```

---

**The prompt used to verify the truthfulness of each proposition against the given source text.**

```
You are a professional content analyst who checks the truthfulness of propositions
    based on a given text.
Read the text provided below and evaluate each proposition for its truthfulness based
    on the content of the text.

**[Requirements]**
1.  **Truthfulness:** For each proposition, determine if it is true or false based
       solely on the information in the given text. Never use external knowledge or make
       guesses.
2.  **Format:** Organize your evaluation in the JSON format, with each proposition
       paired with a reasoning and a boolean indicating its truthfulness.

**[Truthfulness Evaluation Criteria]**

True:
* When the information is explicitly stated in the text
* When it is not directly mentioned but can be necessarily inferred from the context of
     the text

False:
* When the information contradicts the text
* When the information is not supported by the text at all and would require external
    knowledge to judge

Now, please start working with the text below.

# Text

{text}

---

# Propositions
{list of propositions}
```

---

**The prompt used to perform masking-based redaction on a given text according to a specified security policy.**

```
You are a professional content redactor who removes sensitive information from a text
    based on given security policies.
Read the text provided by user and redact(masking) any information that violates the
    provided security policies.
In doing so, you must preserve non-sensitive information as much as possible to
    maintain the text's usefulness and context.

# Policies
{list of policies}

# Format

Reasoning first, then present the final redacted text at the end of your entire
    response, preceded by the phrase '# Redacted text'.

# Text

{text}
```

The prompt used to perform adversarial redaction on a given text according to a specified security policy (Step 1).

```
Analyze the [TEXT TO ANALYZE] below based on the provided [SECURITY POLICY]. Identify
    and list all pieces of sensitive information that violate the policy. For each
    violation, provide the reason. Format your response as a bulleted list.

---
**[SECURITY POLICY]:**
{list of policies}

---
**[TEXT TO ANALYZE]:**
{text}
```

The prompt used to perform adversarial redaction on a given text according to a specified security policy (Step 2).

```
You are a professional content redactor who carefully adapts small parts of the text to
    make it less sensitive.
Review the [TEXT TO PROCESS] below. Find and remove all sensitive information matching
    the contents listed in [INFERRED SENSITIVE INFORMATION].

Your modifications must follow these rules:
1.  **Generalize, do not mask:** Replace sensitive details with general descriptions (e
    .g., "John Doe" becomes "the employee"). Do not use placeholders like "[REDACTED
    ]".
2.  **Minimize changes:** Keep the original text as intact as possible, modifying only
    what is necessary.
3.  **No new information:** Do not add any facts or details that were not in the
    original text.

---
**[INFERRED SENSITIVE INFORMATION]:**
{response from step 1}

---
**[TEXT TO PROCESS]:**
{text}

# Response Format
Reasoning first, then present the final redacted text at the end of your entire
    response, preceded by the phrase '# Redacted text'.
```

