# OpenReview forum: "RedacBench: Can AI Erase Your Secrets?"
_ICLR.cc/2026/Conference — ICLR 2026 Poster_

### Official Review · Reviewer_TFrj · 2025-10-21

**Soundness:** 2
**Presentation:** 3
**Contribution:** 2
**Rating:** 4
**Confidence:** 4

**Summary:**

The authors introduce RedacBench, a novel benchmark for evaluating redaction capabilities. Specifically, through 514 human-written texts and 187 security policies, it measures a model’s ability to redact diverse forms of sensitive information embedded in text. The authors conduct experiments on 9 language models with different redaction strategies, highlighting that effective redaction comes as a tradeoff between security and utility.

**Strengths:**

- The paper addresses a timely problem, is clearly written, and well-structured.
- The introduced dataset is valuable; the use of propositions effectively supports redaction evaluation and allows easy scalability with new policies.
- The playground is useful for exploring and understanding how the benchmark operates.

**Weaknesses:**

- Some redaction strategies are prone to hallucinations, with no mechanism provided to detect or account for them.
- No comparison with related work; RedacBench performance is not contextualized against existing studies.

**Questions:**

- Although the authors acknowledge hallucinations among the method’s limitations, they associate them solely with cases in which the model has been trained on the document to be redacted, and might therefore recall the redacted information.
My primary concern, however, is that hallucinations may occur regardless of whether the model is already familiar with the text. With adversarial redaction, a language model identifies sensitive information and rewrites the text accordingly. Regardless of whether the model has been previously exposed to this sample, a hallucination may still occur and compromise the rewritten text.

    For example, in the first redaction samples reported in Appendix B, we have:

    **Original Text:**

    *“Bob, regarding Patti Sullivan’s contributions to the west desk this year […]”*

    **Adversarial Redaction 1 Text:**

    *“Bob, regarding the employee’s contributions to the team this year […]”*

    However, a model could also redact that information by, for instance, replacing the employee’s name with another one:

    *“Bob, regarding Alice’s contributions to the team this year […]”*

    Although this could be considered secure (as a true negative), it may still affect the utility of the text if the meaning changes. This type of issue is not captured by the Utility Score proposed by the authors.

    My question, therefore, is the following:

    Have you observed this kind of phenomenon with Adversarial Redaction? If so, how frequently does it occur? And how could the Utility Score be modified to account for these types of failures in the redaction process?

- There is no analysis that relates the models’ performance on RedacBench to those reported in other studies. Moreover, in the Related Work section, there is no explanation as to why the cited works cannot be directly compared with the approach introduced by the authors. For instance, a comparison could have been made with the Adaptive PII Mitigation Framework regarding the redaction of specific PII data. Furthermore, for example, it would have been interesting to evaluate the performance of RedacBuster after applying the various redaction techniques proposed in RedacBench, in order to assess, for example, the robustness of the solutions with high Security or Utility Scores.

---

> ### Author Response · Authors · 2025-11-25
>
> Dear Reviewer TFrj,
>
> We sincerely appreciate your review with thoughtful comments. We have carefully considered each of your questions and provide detailed responses below. Please let us know if you have any further questions or concerns.
>
> **[W1, Q1] No Mechanism to Detect or Account for Hallucinations**
>
> Thank you for pointing out that some redaction strategies may introduce hallucinated content, which is not fully captured by the existing utility metric.
>
> To address this, we implemented a hallucination-detection mechanism based on reverse entailment:
>
> 1. We extract propositions from the redacted text.
> 2. We verify whether each proposition is entailed by the original source.
> 3. Any proposition not supported by the source is marked as a hallucination.
>
> This enables us to compute a **hallucination rate**, which complements the existing utility score by penalizing unsupported additions to the text (e.g., changing “Patti” to “Alice”). This extension provides a more complete evaluation by capturing both preservation (recall of true information) and faithfulness (avoidance of hallucinations).
>
> We applied this analysis to the adversarial redaction method using GPT-4.1 and observed a hallucination rate of **3.36%** (150 out of 4,460 instances). These results indicate that language models can introduce unsupported information during redaction. Although the rate is relatively low, it still motivates future research on reducing such errors. We have incorporated these findings, along with corresponding discussion, into the **Appendix C** of the revised manuscript.
>
> ---
>
> **[W2, Q2] Lack of Comparison with Related Work**
>
> Thank you for the opportunity to clarify our contribution in the context of prior work.
>
> While we agree that a comparison with conventional methods (such as NER-based recognition used in adaptive PII mitigation frameworks) is valuable, these methods assume that sensitive information corresponds to identifiable entities in the input text, a limitation that has been well discussed in recent literature [1]. For this reason, rather than benchmarking against techniques whose limitations have already been broadly established, we emphasize two key differences where RedacBench addresses gaps that other frameworks cannot:
>
> **1. Expanded Scope.** RedacBench evaluates complex corporate and government documents where sensitivity is defined by high-level security policies, not fixed categories like names or addresses. Existing benchmarks limited to PII cannot capture these broader, policy-driven notions of sensitive content.
>
> **2. Broader Strategies.** PII masking tools primarily evaluate removal of entity spans. In contrast, our proposition-based framework evaluates whether meaningfully sensitive information is preserved, removed, or generalized.
>
> We have revised the **Related Work** section to clarify why existing entity-based evaluation methods are not suitable baselines and to better articulate the distinct scope and contributions of RedacBench.
>
> ---
>
> **References**
>
> [1] Staab et al. “Beyond Memorization: Violating Privacy Via Inference with Large Language Models” ICLR, 2024.

---

> > ### Author Response · Authors · 2025-11-28
> >
> > Dear Reviewer TFrj,
> >
> > We hope our response has addressed your concerns. We would be grateful if you could share any further feedback or clarify whether the issues have been resolved.
> >
> > Thank you for your time.
> >
> > Best regards,
> > Authors

---

### Official Review · Reviewer_icaH · 2025-10-29

**Soundness:** 3
**Presentation:** 3
**Contribution:** 3
**Rating:** 8
**Confidence:** 3

**Summary:**

The paper proposes a benchmark for evaluating text redaction methods. The benchmark works by using an LLM autograder to determine whether key units of factual information from the unredacted text could be inferred from the redacted text alone. Data is collected from three varied sources (real human data). A human-in-the-loop approach is used, along with an LLM, to extract key factual units from the unredacted texts, along with privacy policies that this information may or may not violate. The autograder shows less than a 2% false-negative rate, making it fairly appropriate for evaluation of information leakage. Evaluations are conducted on frontier API-based LLMs, showing a Pareto frontier on redacted text informativeness and information leakage, with Claude 4 and the Adversarial Redaction (Staab et al, 2025) method performing best.

**Strengths:**

- Very important: The benchmark for redaction methods appears to be reasonably constructed, and I believe it will be useful for the privacy community.
- Important: The shift toward inference-based measures of privacy is very valuable. Using carefully validated LLM graders seems like the right approach for this.
- Important: A severe tradeoff between utility and security of current SOTA models/methods suggests that there is significant room for improvement of redaction methods on the benchmark.
- Of some importance: The web app also provided alongside the dataset is an interesting artifact and will hopefully allow for researchers to quickly prototype new ideas.

**Weaknesses:**

- Important: I’m struggling to get a sense of what optimal performance might look like on the dataset. Do you have examples that can be perfectly redacted by a human? It would help to know what the ceiling performance is on the benchmark and how far models are from this performance. Otherwise it may not be clear to the community how long to work on this benchmark, when it is saturated, etc.
- Important: I couldn’t see any detail to who the humans were in the human-in-the-loop data construction process. More detail on this is important.
- Of some importance: I was slightly concerned by two experiment design choices: using a weaker model grader than the redaction model, in some cases, (gpt-5 vs. gpt-4.1-mini), and the lack of ablation across redaction model and grader model families (grader models may unfairly favor their own redaction models efforts).

**Questions:**

Please feel free to respond to the questions in the weaknesses section.

---

> ### Author Response · Authors · 2025-11-25
>
> Dear Reviewer icaH,
>
> We sincerely appreciate your review with thoughtful comments. We have carefully considered each of your questions and provide detailed responses below. Please let us know if you have any further questions or concerns.
>
> **[W1] Missing Human Baseline and Ceiling Performance Analysis**
>
> Thank you for raising the importance of establishing a performance ceiling. We agree that understanding what optimal redaction looks like is essential for interpreting progress on RedacBench. To address this, we manually redacted the dataset, optimizing for both security (complete removal of sensitive propositions) and utility (preservation of non-sensitive content).
>
> Our findings show that manual redaction performs substantially **better than all evaluated redaction methods**. This large gap shows that the benchmark is far from saturated and that significant headroom remains for improving automated redaction systems.
>
> | Version  | Security | Utility |
> | --------- | -------- | ------- |
> | Optimal-1 | 62.8     | 85.2    |
> | Optimal-2 | 69.8     | 66      |
> | Optimal-3 | 72.6     | 57.5    |
>
> [Utility–security trade-off graph with a human baseline](https://redacbench.vercel.app/Optimal.png)
>
> We have added this human baseline to **Figure 3a** and the detail to **Appendix F** in the revised manuscript to provide a clear reference point for ceiling performance.
>
> ---
>
> **[W2] Insufficient Detail on Human Annotators**
>
> We thank the reviewer for pointing out the need for more details regarding the human annotators involved in data construction. We agree that this information is important for assessing data quality and reliability.
>
> In the revised paper (**Section 2.3**), we now provide details about the two annotators involved:
>
> 1. One author with research experience in AI privacy and security.
> 2. One external professional with over five years of experience working at a national university and an English-speaking global consulting firm.
>
> Both annotators were fully briefed on the data synthesis pipeline and their tasks. Disagreements were resolved through discussion until consensus was reached to ensure annotation consistency.
>
> ---
>
> **[W3] Model-Grader Mismatch and Lack of Cross-Family Ablation**
>
> Thank you for highlighting concerns about using a smaller evaluator model and the possibility of family-specific grading bias. To address this, we conducted an ablation in which GPT-5, GPT-4.1-nano, Gemini-2.5-Flash, and Claude-Sonnet-4.5 were each used as evaluators on the same set of redacted outputs produced by GPT-4.1. Our findings are as follows:
>
> **1. Impact of Model Capability on “Strictness” and Security Scores.**
> We observed that larger models tended to be significantly “stricter” in their criteria for judging whether a proposition could be inferred. Consequently, these excessively capable models produced inflated Security Scores and deflated Utility Score, as they were more likely to classify a proposition as “removed” when the text was merely altered rather than fully sanitized.
>
> **2. Consistency of Rankings (Addressing Bias).**
> Regarding the concern about self-preference bias, our ablation results showed that the relative rankings of the redaction methods remained consistent across all evaluators. Whether the grader was from the GPT, Gemini, or Claude family, the comparative performance between redaction strategies did not change. This indicates that the choice of evaluator did not introduce a bias significant enough to alter the paper’s conclusions.
>
> | Evaluation Model | Masking (Security) | Masking (Utility) | AR (iter 1) (Security) | AR (iter 1) (Utility) | AR (iter 2) (Security) | AR (iter 2) (Utility) |
> | :--- | :---: | :---: | :---: | :---: | :---: | :---: |
> | gpt-5 | 86.6 | 36.8 | 92.6 | 29.4 | 96.6 | 18.6 |
> | gemini-2.5-flash | 73.9 | 48.6 | 83.4 | 39.8 | 90.1 | 28.2 |
> | claude-sonnet-4.5 | 78.2 | 41.5 | 85.7 | 35.6 | 91.5 | 25.1 |
> | gpt-4.1-mini | 36.4 | 82.0 | 68.2 | 55.1 | 77.0 | 44.4 |
> | gpt-4.1-nano | 31.8 | 86.4 | 43.2 | 80.9 | 48.8 | 77.6 |
>
> [Utility–security trade-off graph of evaluation model ablation](https://redacbench.vercel.app/Eval.png)
>
> Based on these experiments, we determined that **GPT-4.1-mini strikes the optimal balance** in terms of strictness for this specific task. It avoids the over-strict evidence thresholds of larger models while maintaining sufficient reasoning capability to detect leaked information. While larger models could potentially be effective with prompt engineering to lower their evidence thresholds, GPT-4.1-mini proved to be the most reliable and balanced evaluator for the current benchmark setup. We have added the results to **Appendix G**.

---

> > ### Comment · Reviewer_icaH · 2025-11-25
> >
> > Thanks for the extra experiments! This raises my confidence that the paper will be a solid contribution to the community. I'll keep my overall score but increase my confidence rating.

---

### Official Review · Reviewer_Bd43 · 2025-10-31

**Soundness:** 3
**Presentation:** 3
**Contribution:** 2
**Rating:** 4
**Confidence:** 3

**Summary:**

This paper makes an important contribution by introducing RedacBench, a comprehensive benchmark for the nuanced task of policy-based text redaction. Its core strengths lie in the novel proposition-based evaluation framework and the high-quality, diverse dataset constructed using a human-in-the-loop methodology. The experimental results clearly demonstrate the critical trade-off between security and utility, providing strong baselines for future work. However, the evaluation framework's reliance on an LLM (GPT-4.1-mini) as an automated evaluator introduces a potential risk of data contamination. Furthermore, the experiments primarily rely on closed-source APIs, which raises significant security and privacy concerns for real-world deployment.

**Strengths:**

- The work moves beyond simple PII detection, formulating a more realistic task of context-sensitive redaction based on specific security policies. The inclusion of 187 multi-layered policies, spanning from granular details to high-level abstract concepts, aligns well with practical requirements.
- The Security Score and Utility Score offer a clear, quantitative, and interpretable method for measuring the inherent trade-off in redaction, which is effectively visualized in the results. The authors establish a reasonable baseline for the evaluator's reliability by testing its false negative rate on known true propositions, achieving a low error rate of 1.45%.
- The benchmark dataset is constructed from diverse and challenging sources, including texts from individual, corporate (Enron), and government (Hillary Clinton) domains. This diversity ensures models are tested across a wide range of real-world scenarios. The public release of the benchmark and the accompanying interactive web playground (Appendix A) is a valuable contribution to the community, lowering the barrier for future research and enabling others to build upon this work.

**Weaknesses:**

- The evaluation framework relies on GPT-4.1-mini as an automated judge, which introduces a risk of "recall" from pre-training data contamination. Even if information is successfully redacted, the evaluator might incorrectly assess it as "preserved" (a false positive), thus artificially deflating the Security Score. The validation only checks the false negative rate and fails to assess the more critical false positive rate, posing a substantial threat to the validity of the reported scores.
- The experiments (Table 3) heavily rely on closed-source models (e.g., GPT-5, GPT-4.1, Claude-Sonnet-4). Sending sensitive text to third-party APIs is operationally infeasible in real-world applications and presents a significant data leakage risk. The paper fails to evaluate the performance of state-of-the-art, locally-deployable open-source models on RedacBench, leaving a gap for organizations that cannot use external APIs to assess practical trade-offs.
- The framework treats every proposition with equal weight, ignoring the differential risk associated with various policies. In a practical scenario, failing to redact a macro-level violation (e.g., "Strategic business plan") is far more severe than a micro-level one (e.g., "Instructor names"). The simple binary sensitive/non-sensitive classification and unweighted scoring risk evaluation distortion. This represents a missed opportunity to leverage the multi-layered policy information (Table 1) for a risk-weighted, granular metric design that would better reflect real-world priorities.

**Questions:**

- The evaluation framework relies on GPT-4.1-mini as an automated judge, which introduces a risk of "recall" from pre-training data contamination. Even if information is successfully redacted, the evaluator might incorrectly assess it as "preserved" (a false positive), thus artificially deflating the Security Score. The validation only checks the false negative rate and fails to assess the more critical false positive rate, posing a substantial threat to the validity of the reported scores.
- The experiments (Table 3) heavily rely on closed-source models (e.g., GPT-5, GPT-4.1, Claude-Sonnet-4). Sending sensitive text to third-party APIs is operationally infeasible in real-world applications and presents a significant data leakage risk. The paper fails to evaluate the performance of state-of-the-art, locally-deployable open-source models on RedacBench, leaving a gap for organizations that cannot use external APIs to assess practical trade-offs.
- The framework treats every proposition with equal weight, ignoring the differential risk associated with various policies. In a practical scenario, failing to redact a macro-level violation (e.g., "Strategic business plan") is far more severe than a micro-level one (e.g., "Instructor names"). The simple binary sensitive/non-sensitive classification and unweighted scoring risk evaluation distortion. This represents a missed opportunity to leverage the multi-layered policy information (Table 1) for a risk-weighted, granular metric design that would better reflect real-world priorities.

---

> ### Author Response · Authors · 2025-11-25
>
> Dear Reviewer Bd43,
>
> We sincerely appreciate your review with thoughtful comments. We have carefully considered each of your questions and provide detailed responses below. Please let us know if you have any further questions or concerns.
>
> **[W1] Evaluator Contamination and False Positive Risk**
>
> We appreciate the reviewer highlighting the importance of evaluating the false positive (FP) rate, where the judge incorrectly concludes that redacted information is still present. To directly measure this, we defined the FP rate as the proportion of propositions recognized as true even after all supplementary context had been removed.
>
> Upon evaluating 8,053 propositions, we observed that GPT-4.1-mini produced 211 false positives, corresponding to a false positive rate of approximately **2.62%**. We view this result as supporting two important points:
>
> **1. Conservative Security Estimates.** A FP rate of 2.62% implies that the Security Score may be slightly deflated, meaning the true redaction performance of the models is likely somewhat better than reported. Our benchmark therefore errs on the side of conservatism rather than overstating safety.
>
> **2. Preservation of Relative Comparisons.** Because this evaluator bias is uniform across all models and settings, relative rankings and comparative conclusions remain unaffected.
>
> We incorporated this analysis in our revised paper (**Section 3.2**) to provide a more complete characterization of evaluator reliability.
>
> ---
>
> **[W2] Dependence on Closed-Source Models**
>
> We thank the reviewer for pointing out the limitations of relying solely on closed-source API models. We agree that many real-world applications require on-premise deployment and cannot share sensitive data with external providers for security reasons. To address this, we added state-of-the-art open-source models to our evaluation. Specifically, we evaluated **Qwen3-8B** and **Qwen3-4B-Instruct-2507** under the same settings as the proprietary models. The results are summarized in the table below.
>
> | Model | Masking Security | Masking Utility | AR (iter 1) Security | AR (iter 1) Utility | AR (iter 2) Security | AR (iter 2) Utility |
> | :--- | :---: | :---: | :---: | :---: | :---: | :---: |
> | **qwen3-8b** | 37.1 | 79.3 | 46.5 | 75.2 | 57.4 | 64.2 |
> | **qwen3-4b-2507** | 51.6 | 72.8 | 63.5 | 59.1 | 75.8 | 44.4 |
>
> For Qwen3-8B, the security score was slightly lower than that of the state-of-the-art proprietary models. Nevertheless, we observed that smaller yet more recent models can achieve competitive results when paired with more effective redaction strategies. For example, the recent Qwen3-4B-Instruct-2507 model achieved performance between GPT-4.1 and GPT-4.1-mini, demonstrating that open-source models can substantially improve redaction quality when incorporating the latest techniques. Moreover, prior work [1] on distillation-based anonymization suggests that the remaining performance limitations of small models can be further mitigated.
>
> These results (now included in **Section 3.2, Table 3 and Figure 2(a)**) provide useful baselines for open-weight deployments and allow organizations to assess the security-utility trade-offs of open models relative to proprietary ones.

---

> ### Author Response · Authors · 2025-11-25
>
> **[W3] Uniform Weighting Ignores Differential Risk**
>
> We thank the reviewer for highlighting the importance of accounting for differential risk across policies. We agree that macro-level violations (e.g., strategic business plans) can carry far greater consequences than micro-level ones (e.g., instructor names), and a robust benchmark should reflect this.
>
> To address this, we performed an additional analysis using a risk-weighted metric as follows:
>
> 1. **Risk Scoring.** Each of the 187 policies is assigned a severity score from 1 to 5, reflecting the practical impact of a violation.
> 2. **Weighted Security Score.** Model performance is recalculated so that redacting high-severity policies contributes more to the final score than lower-severity ones.
>
> The weighted security score results are summarized in the table below.
>
> | Model | Masking (non-weighted) | Masking (weighted) | AR (iter 1) (non-weighted) | AR (iter 1) (weighted) | AR (iter 2) (non-weighted) | AR (iter 2) (weighted) |
> | :--- | :---: | :---: | :---: | :---: | :---: | :---: |
> | gpt-5 | 38.9 | 38.3 | 72.3 | 71.7 | 77.1 | 76.6 |
> | gpt-5-mini | 41.8 | 41.3 | 63.4 | 62.8 | 80.9 | 80.5 |
> | gpt-5-nano | 38.5 | 37.5 | 51.9 | 50.9 | 58.2 | 57.4 |
> | gpt-4.1 | 36.4 | 35.7 | 68.2 | 67.6 | 77.0 | 76.5 |
> | gpt-4.1-mini | 37.2 | 36.3 | 53.7 | 52.8 | 60.2 | 59.4 |
> | gpt-4.1-nano | 40.7 | 39.9 | 64.1 | 63.5 | 61.7 | 60.9 |
> | gemini-2.5-flash | 43.9 | 43.2 | 56.2 | 55.4 | 61.7 | 60.9 |
> | gemini-2.5-flash-lite | 35.9 | 34.9 | 52.2 | 51.3 | 60.2 | 59.4 |
> | claude-sonnet-4 | 44.6 | 43.6 | 59.5 | 58.6 | 68.5 | 67.8 |
> | qwen3-8b | 37.1 | 36.1 | 46.5 | 45.8 | 57.4 | 56.9 |
> | qwen3-4b-2507 | 51.6 | 50.7 | 63.5 | 62.7 | 75.8 | 75.4 |
>
> After applying the policy-specific weights, the overall security scores decreased. This suggests that high-severity security policies include requirements that are inherently more difficult to satisfy. We have included the detailed methodology and results of the risk-weighted evaluation in **Appendix E**.
>
> We also note that this perspective suggests a valuable avenue for future work: dynamically adjusting redaction strategies based on policy severity to maximize safety while preserving utility. This discussion has been added to the revised paper.
>
> ---
>
> **References**
>
> [1] Kim et al. “Self-Refining Language Model Anonymizers via Adversarial Distillation” NeurIPS, 2025.

---

> ### Author Response · Authors · 2025-11-28
>
> Dear Reviewer Bd43,
>
> We hope our response has addressed your concerns. We would be grateful if you could share any further feedback or clarify whether the issues have been resolved.
>
> Thank you for your time.
>
> Best regards,
> Authors

---

### Official Review · Reviewer_iL3S · 2025-11-06

**Soundness:** 3
**Presentation:** 3
**Contribution:** 2
**Rating:** 4
**Confidence:** 3

**Summary:**

- This work introduces RedacBench, a benchmark for evaluating LLM-based text redaction.
- It provides a standardized framework to quantitatively assess the trade-off between security and utility, offering an objective basis for comparing diverse redaction techniques.
- RedacBench lays the groundwork for trustworthy AI systems capable of securely managing and erasing sensitive information.

**Strengths:**

- The paper introduces a fine-grained, proposition-level analysis that captures semantic inferability of information, enabling more rigorous and interpretable quantification of redaction effectiveness than surface-level token or entity matching.
- Defines complementary metrics — Security Score (true negative rate for sensitive information) and Utility Score (true positive rate for non-sensitive information) — allowing quantitative assessment of both privacy protection and information preservation.
- Curates multiple human-written documents and 187 distinct policies, annotated into over 8,000 atomic propositions. This design supports fine-grained, cross-domain testing and reflects realistic privacy constraints
- Evaluates multiple redaction strategies (e.g., masking, rewriting, policy-based prompting) across various state-of-the-art LLMs, establishing meaningful baselines and empirically demonstrating the trade-off between model capability, security, and text utility.

**Weaknesses:**

- The benchmark evaluates models in a controlled, static setting, but it does not test interactive or context-evolving scenarios where redaction systems must operate dynamically (e.g., during live conversations or document editing).
- Although the paper positions redaction as a privacy defense, it does not directly compare or align its evaluation metrics with other privacy frameworks like unlearning, membership inference resistance.
- The current benchmark quantifies empirical removal, not privacy guarantees.
- The binary classification of propositions as “removed” or “preserved” may not capture partial or implicit leakage — cases where sensitive information is paraphrased, entailed, or inferable through context reconstruction.
- The paper focuses on improving redaction accuracy but does not address the ethical and practical risks of over-redaction, such as erasing legitimate public-interest information or introducing bias in released datasets.

**Questions:**

- Can your evaluation account for implicit leakage — cases where sensitive information is not directly stated but can still be inferred from contextual cues or correlated details?
- How does RedacBench relate to or complement benchmarks used for evaluating unlearning, differential privacy, or content filtering methods?
- The paper defines utility as the preservation of non-sensitive propositions. Have you considered complementary measures of semantic consistency?

---

> ### Author Response · Authors · 2025-11-25
>
> Dear Reviewer iL3S,
>
> We sincerely appreciate your review with thoughtful comments. We have carefully considered each of your questions and provide detailed responses below. Please let us know if you have any further questions or concerns.
>
> **[W1] Static Evaluation Does Not Reflect Dynamic Real-World Use**
>
> We thank the reviewer for suggesting the evaluation of dynamic scenarios. While our original submission used a static evaluation as a baseline, real-world redaction often occurs interactively. To address this, we conducted an additional experiment simulating a context-evolving scenario:
>
> 1. Each text $T$ was split into $k$ sequential chunks $(t_1, t_2, \dots, t_k)$, roughly corresponding to sentences or short paragraphs.
> 2. The model acted as a multi-turn “readaction assistant”. At turn $n$, it received a chunk $t_n$ along with the conversation history and produced a redacted output $r_n$.
> 3. Sequential outputs were concatenated $(r_1 \oplus r_2 \oplus \dots \oplus r_k)$ and evaluated with the standard proposition-based RedacBench metrics.
>
> This experiment highlights the challenge of missing “forward context” in dynamic scenarios. Using GPT-4.1-nano for adversarial redaction, we observed a Security Score of **55.2** and a Utility Score of **60.9**, representing a **notable drop in security** compared to the static baseline (Security Score of 64.1 and a Utility Score of 52.6), as the model fails to identify sensitive information that requires context found only in later segments of the text.
>
> We have added this analysis to the **Appendix B** to illustrate RedacBench’s applicability to dynamic, interactive workflows.
>
> ---
>
> **[W2, Q2] Alignment with Other Privacy Frameworks**
>
> We thank the reviewer for this insightful comment. We agree that clarifying RedacBench’s relationship to broader privacy frameworks strengthens the paper. We have updated the **Related Work** section to make these distinctions explicit:
>
> **1. Distinct Scope (Inference vs. Training).** Frameworks such as machine unlearning focus on sanitizing model knowledge (preventing memorization or regurgitation of training data). RedacBench instead targets inference-time inputs and outputs, which is critical for applications like AI assistants where models encounter new sensitive data not present in training.
>
> **2. Complementary Defense.** Even models with perfect unlearning or DP protections require robust redaction to safely handle sensitive user inputs, making RedacBench a complementary, inference-time safeguard.
>
> **3. Metric Alignment.** Our proposition-based evaluation measures the removal of sensitive information while preserving context, providing a fine-grained view of privacy that existing metrics cannot fully capture.
>
> ---
>
> **[W3, Q2] Empirical Removal vs. Formal Privacy Guarantees**
>
> We agree that RedacBench is designed to measure contextual integrity and policy compliance in unstructured text, rather than providing formal privacy guarantees. However, we believe this empirical approach is critical for the current state of LLM deployment for the following reasons, which we have clarified in the Limitations paragraph of the **Discussion** section in the revised manuscript:
>
> **1. Formal guarantees are difficult for text.** Techniques such as differential privacy work well for structured data, but applying them to unstructured text often severely degrades fluency and meaning. In practical settings, such as legal documents or corporate emails, ensuring policy compliance is more relevant than statistical indistinguishability. RedacBench directly tests whether sensitive facts are semantically removed while preserving the rest of the text.
>
> **2. Simulating realistic inference attacks.** Empirical removal is measured by using a strong LLM as an adversary attempting to infer sensitive facts from the redacted output. If even this model fails, the information is effectively inaccessible to state-of-the-art inference attacks. While this is not a cryptographic guarantee, it provides a practical lower bound on security against real-world threats.

---

> ### Author Response · Authors · 2025-11-25
>
> **[W4, Q1] Limitations of Binary Classification Measures**
>
> We thank the reviewer for raising this important privacy concern. Detecting implicit leakage, where sensitive information is paraphrased, entailed, or recoverable through contextual reconstruction, is a core design principle of RedacBench. To address this, our evaluation framework includes two mechanisms that go beyond surface-level or keyword-based matching:
>
> **1. Semantic Proposition Extraction (Section 2.3).** Rather than segmenting text into literal fragments, we generate fine-grained semantic propositions that capture implicit information present in the original text.
>
> **2. Inference-Based Evaluation (Section 3.2).** Instead of checking for lexical overlap, we use an LLM evaluator instructed to decide if a proposition remains inferable from the redacted text, regardless of paraphrasing, rewording, or indirect cues. This explicitly tests for partial or implicit leakage.
>
> Consider the sample from Table 2,
>
> * Original Text: *“If there is a creative way to structure the deal... One idea that has been mentioned is to obtain a ‘forward commitment’ in order to reduce the equity required.”*
> * Proposition 4: *“A financing structure using a ‘forward commitment’ is being considered to reduce required equity.”*
>
> Even if a model simply redacts the term “forward commitment” or rephrases the sentence, as long as it retains the core description of the mechanism (e.g., “obtaining a promise to fund later to lower upfront costs”), the evaluator will determine that Proposition 4 is still inferable from the context.
>
> ---
>
> **[W5] Over-Redaction Risks and Ethical Implications Not Addressed**
>
> We thank the reviewer for raising the important ethical and practical risks associated with over-redaction, including the removal of legitimate public-interest information and the potential introduction of bias. We have addressed these concerns in two ways:
>
> **1. Measuring Over-Redaction with the Utility Score.** We agree that excessive removal of valid information is a critical failure mode. RedacBench explicitly penalizes this behavior: the Utility Score measures the preservation of non-sensitive propositions, ensuring that systems cannot achieve high security simply by deleting large portions of text. This provides a quantitative safeguard against over-redaction.
>
> **2. New Analysis on Bias and Hallucination.** We share the concern that redaction, particularly generative rewriting, may introduce bias or distortions. In our view, this risk primarily manifests through hallucinations. To address it, we implemented a hallucination-detection mechanism based on reverse entailment. Here, we (a) extract propositions from the redacted text, (b) verify whether each proposition is checked against the original source, and (c) unsupported propositions are marked as hallucinations.
>
> Applying this analysis to the adversarial redaction with GPT-4.1, we observed a hallucination rate of **3.36%** (150 out of 4,460 propositions). While relatively low, this confirms that LLMs can introduce fabricated content. We have incorporated this analysis into the revised **Appendix C**.
>
> ---
>
> **[Q3] Lack of Semantic Consistency Measures in Utility Definition**
>
> We appreciate the reviewer’s feedback regarding utility evaluation. While our proposition-based metric effectively measures the retention of atomic facts, we acknowledge the need to assess broader semantic consistency. To address this, we have incorporated an LLM-as-a-judge framework to evaluate utility across three dimensions: Semantic Similarity (overall meaning), Readability (fluency and coherence), and Hallucinations (factual fabrication).
>
> Following established practices ([1], [2]), we applied this framework to texts redacted by GPT-4.1 using three distinct methods. Our analysis yielded the following observations:
>
> * **Semantic Similarity.** Scores appeared less strict than the utility score in RedacBench.
> * **Readability.** The masking-based approach showed a significant decline in readability, as expected.
> * **Hallucinations.** Scores reached near-maximum values across all methods, exhibiting a ceiling effect and limited discriminative power.
>
> | Redaction Method | RedacBench | Similarity | Readability | Hallucinations |
> | :--- | :--- | :--- | :--- | :--- |
> | Masking | 82.0 | 77.5 | 89.1 | 99.8 |
> | AR (iter 1) | 55.1 | 80.2 | 94.8 | 99.2 |
> | AR (iter 2) | 44.4 | 72.8 | 93.2 | 99.4 |
>
> These results underscore the trade-offs between meaning preservation, readability, and factual integrity. We believe this multi-dimensional analysis provides a more comprehensive evaluation of the security-utility trade-off. This result has been added to **Appendix D** of the revised paper.
>
> ---
>
> **References**
>
> [1] Staab et al. “Large Language Models are Advanced Anonymizers” ICLR, 2025.
>
> [2] Kim et al. “Self-Refining Language Model Anonymizers via Adversarial Distillation” NeurIPS, 2025.

---

> ### Author Response · Authors · 2025-11-28
>
> Dear Reviewer iL3S,
>
> We hope our response has addressed your concerns. We would be grateful if you could share any further feedback or clarify whether the issues have been resolved.
>
> Thank you for your time.
>
> Best regards,
> Authors

---

### Author Response · Authors · 2025-11-25
**Response to All Reviewers**

We thank all reviewers for their insightful and constructive feedback. We are encouraged by the reviewers’ recognition of the importance of RedacBench in addressing the gap between static PII masking and dynamic, policy-driven redaction.

Based on the reviewers’ suggestions, we have conducted extensive additional experiments to strengthen the paper’s empirical rigor and broaden its scope. Below is a summary of the major updates made to the manuscript.

**1. Expanded Evaluation Scenarios.**
To address concerns regarding the realism and applicability of the benchmark, we introduced two key experimental expansions:

* Dynamic Context (Reviewer iL3S): We simulated a multi-turn, sequential redaction scenario. Results show that missing “forward context” causes a notable drop in security (Security Score drops from 64.1 to 55.2), highlighting the difficulty of real-time redaction.
* Open-Source Models (Reviewer Bd43): We added evaluations for Qwen3-8B and Qwen3-4B-Instruct-2507. The results demonstrate that while smaller open-source models trail proprietary ones, they offer viable baselines for on-premise deployment.

**2. Robustness of the Metric and Evaluator.**
We performed rigorous validation of our LLM-as-a-judge framework to ensure reliability:

* Optimal Baseline (Reviewer icaH): We established a “gold standard” via manual redaction. The significant gap between human performance and state-of-the-art models confirms that RedacBench remains a challenging, unsaturated benchmark.
* Evaluator Ablation & Bias (Reviewer icaH, Bd43): We tested multiple evaluator models (GPT-5, Claude, Gemini) and measured the False Positive rate (2.62%). Results confirm that our choice of evaluator (GPT-4.1-mini) provides a balanced and consistent ranking of redaction methods without introducing family-specific bias.

**3. Enhanced Utility and Risk Analysis.**
We refined how we measure the “quality” of redaction beyond simple proposition retention:

* Hallucination Detection (Reviewer iL3S, TFrj): We implemented a reverse-entailment check to penalize generative models that fabricate information (e.g., changing names rather than removing them).
* Multi-Dimensional Utility (Reviewer iL3S): We added assessments for Semantic Similarity and Readability to provide a holistic view of text quality.
* Risk-Weighted Scoring (Reviewer Bd43): We introduced a severity-weighted metric, revealing that models struggle more with high-stakes security policies.

**4. Conceptual Clarifications.**
We have revised the **Related Work** and **Discussion** sections to clearly distinguish RedacBench from machine unlearning (inference vs. training) and standard NER-based anonymization (policy compliance vs. entity masking).

We have incorporated these analyses into the main text and **Appendices B through G**. We believe these revisions significantly strengthen the paper and thank the reviewers for guiding these improvements.

Thank you,

Authors

---

### Meta-Review · Area_Chair_4q6U · 2026-01-09

**Summary:**

- The benchmark for redaction methods is timely.
- The shift toward inference-based measures of privacy is very valuable.
- Using carefully validated LLM graders seems like the right approach for this.
- A severe tradeoff between utility and security of current SOTA models/methods suggests that there is significant room for improvement of redaction methods on the benchmark.

**Reviewer Concerns:**

The following reviewer concerns have been addressed in the rebuttal.

- the lack of realistic, interactive scenarios.
- measures for semantic consistency and over-redaction risks are missing.
- alignment with Unlearning/DP and clarification on implicit leakage.
- concerns about "recall" contamination leading to False Positives (FP), which could undermine the Security Score.
- the reliance on proprietary APIs (GPT-4, Claude) and real-world data leakage risks.

**Reviewer Scores:**

Reviewer iL3S would have increased their score.
Reviewer Bd43 would have increased their score.
Reviewer icaH would kept the score.
 Reviewer TFrj would have increased their score.

---

### Decision · Program_Chairs · 2026-01-26

Accept (Poster)